# Prediction of Motor Recovery in Patients with Basal Ganglia Hemorrhage Using Diffusion Tensor Imaging

**DOI:** 10.3390/jcm9051304

**Published:** 2020-05-01

**Authors:** Yu-Sun Min, Kyung Eun Jang, Eunhee Park, Ae-Ryoung Kim, Min-Gu Kang, Youn-Soo Cheong, Ju-Hyun Kim, Seung-Hwan Jung, Jaechan Park, Tae-Du Jung

**Affiliations:** 1Department of Rehabilitation Medicine, School of Medicine, Kyungpook National University, Daegu 41944, Korea; ssuni119@gmail.com (Y.-S.M.); ehmdpark@naver.com (E.P.); ryoung20@hanmail.net (A.-R.K.); 2Department of Rehabilitation Medicine, Kyungpook National University Hospital, Daegu 41944, Korea; kjoohyun88@gmail.com (J.-H.K.); pyromyth@naver.com (S.-H.J.); 3Department of Biomedical Engineering, Seoul National University College of Medicine, Seoul 03080, Korea; 4Department of Medical and Biomedical Engineering, Kyungpook National University, Daegu 41944, Korea; soulmate907@naver.com; 5Department of Physical Medicine and Rehabilitation, Dong-A University College of Medicine, Busan 49201, Korea; kangmingu.ryan@gmail.com; 6Department of Rehabilitation Medicine, Maryknoll Hospital, Busan 48972, Korea; dbstnone@naver.com; 7Department of Neurosurgery, School of Medicine, Kyungpook National University, Daegu 41944, Korea; 8Biomedical Research Institute, School of Medicine, Kyungpook National University, Daegu 41944, Korea

**Keywords:** diffusion tensor imaging, magnetic resonance imaging, recovery, rehabilitation, stroke

## Abstract

Predicting prognosis in patients with basal ganglia hemorrhage is difficult. This study aimed to investigate the usefulness of diffusion tensor imaging in predicting motor outcome after basal ganglia hemorrhage. A total of 12 patients with putaminal hemorrhage were included in the study (aged 50 ± 12 years), 8 patients were male (aged 46 ± 11 years) and 4 were female (aged 59 ± 9 years). We performed diffusion tensor imaging and measured clinical outcome at baseline (pre) and 3 weeks (post1), 3 months (post2), and 6 months (post3) after the initial treatment. In the affected side of the brain, the mean fractional anisotropy (FA) value on pons was significantly higher in the good outcome group than that in the poor outcome group at pre (*p* = 0.004) and post3 (*p* = 0.025). Pearson correlation analysis showed that mean FA value at pre significantly correlated with the sum of the Brunnstrom motor recovery stage scores at post3 (*R* = 0.8, *p* = 0.002). Change in the FA ratio on diffusion tractography can predict motor recovery after hemorrhagic stroke.

## 1. Introduction

Prediction of prognosis in patients with hemorrhagic stroke is difficult but important when planning and designing an individual patient’s rehabilitation strategy [1]. Many studies have used clinical features and neurophysiological findings to predict prognosis, such as the Predict Recovery Potential (PREP) model, which is based on the proportional recovery theory [2,3]. However, most of these studies investigated patients with ischemic stroke rather than hemorrhagic stroke [4,5]. The pathogenesis and recovery mechanisms of hemorrhagic and ischemic stroke are different [6,7,8,9,10]; therefore, models developed for ischemic stroke do not necessarily apply to hemorrhagic stroke patients [11].

The most common mechanism of neuronal injury after hemorrhagic stroke is Wallerian degeneration, which primarily occurs in the corticospinal tract (CST) both adjacent and distant to the hematoma site [11]. Growing evidence shows the potential of diffusion tensor imaging (DTI) as a noninvasive magnetic resonance imaging (MRI) technique for in vivo quantification of microstructural damage to white matter (WM) tracts following stroke [12]. Conventional noninvasive imaging with conventional MRI is not sensitive enough to obtain information about the extent of damage to guide management strategies more effectively [11]. Essentially the main parameter that is measured in DTI is the degree of fractional anisotropy (FA) in a given voxel of the precessing proton and its Eigen vector in an ellipsoid domain [11]. Recently, new diffusion measures—quantitative anisotropy (QA)—have been introduced to the field of DTI for the analysis of diffusion properties of white matter [13,14]. QA represent how much water diffuses (i.e., density) in a specific/restricted direction and in an isotropic fashion (i.e., total isotropic component), respectively. In addition, the difference between QA and FA pertains to the fact that QA is a measure of water diffusion along each fiber orientation, whereas FA is defined for each voxel. Compared to FA, QA is also reported to have a lower susceptibility to partial volume effects of crossing fibers, free-water diffusion in ventricles, and non-diffusive particles [14]. Despite QA’s methodological superiority, FA is the most basic parameter that is intuitively compared the degree of white matter integrity by comparing the ratio of bilateral hemispheres. Furthermore, because the value of FA has been extensively used in previous studies, it is more appropriate to compare the results between studies.

Byblow et al. showed the value of DTI in the subacute phase in identifying which initially severe patients have some potential for recovery and which do not. Other predictors usually have only a moderate predictive effect such as infarct volume, age, gender, dose of rehabilitation treatments, comorbidities, and stroke subtype [2,15,16,17,18]. DTI is increasingly demonstrating greater accuracy in detecting microstructural damage related to Wallerian degeneration after acute stroke [19,20].

Several studies have investigated its utility in predicting prognosis after intracranial hemorrhage (ICH) [21,22,23,24,25,26]. Kusano et al. demonstrated that relative fractional anisotropy (rFA) can reflect prognosis [21]. Other researchers have also shown similar results. Kuzu et al. showed the FA values of the cerebral peduncle on the pathological side in patients with ICH on day 3 can predict motor function on day 90 [22]. Wang et al. showed that the use of DTI during the early stages of ICH may predict motor outcomes at 6 months after ICH. Moreover, as compared to use of DTI within 3 days of ICH onset, the application of DTI at 2 weeks after ICH could more accurately predict the motor outcomes and daily living activities of patients [23]. Koyama et al. developed a model in which FA values assessed by DTI at 2 weeks correlated well with motor outcome at 1 month after onset [26]. However, most of these studies were cross-sectional in design, and only two provided a longitudinal evaluation [24,27]. Wang et al. demonstrated a longitudinal change of rFA after ICH and measured motor outcome with manual muscle testing (MMT), which is too broad to reflect the integrity of the CST [27]. In addition, patients were followed up for only 2 weeks after onset in their study. Although Ma et al. investigated change in FA in acute and chronic hemorrhagic stroke over a 3 month period, longer follow-up is needed because prognosis in patients with a severely impaired CST is difficult to predict [24].

Therefore, this study aimed to investigate the value of diffusion tensor imaging in predicting motor recovery after basal ganglia hemorrhage. The hypothesis of the study is that the initial FA value of DTI imaging after stroke onset would reflect the motor outcome of recovery.

## 2. Materials and Methods

### 2.1. Participants

The study population consisted of 12 right-handed patients with putaminal cerebral hemorrhage who were treated from February 2016 to August 2017. The inclusion criteria were as follows: (1) supratentorial ICH involving the posterior limb of the internal capsule, (2) hematoma volume under 60 mL, (3) age over 20, (4) motor deficit less than Gr 2 by manual muscle test present at the time of admission. The exclusion criteria were as follows: (1) history of previous stroke, (2) any other neurological condition, (3) unstable medical condition pathology, (4) follow-up loss. DTI was performed within 1 day after stroke onset. Patients were followed up at 3 weeks, 3 months, and 6 months after the initial treatment. Outcome at 6 months was classified as good or poor according to the Brunnstrom motor recovery stage (BMS). This study was approved by the institutional review board (IRB) of Kyungpook National University Hospital (2016-08-009).

### 2.2. Evaluation of Clinical Outcome

Motor recovery was the primary outcome and was measured using the BMS of the affected hand. The BMS grades function on a scale of 1–6, with 1 indicating flaccid paralysis and 6 indicating normal or near-normal strength through the complete range of motion. Patients were then stratified into good (hand BMS 0–2) and poor (hand BMS 3–6) motor function groups. Secondary outcome measurements included the Glasgow Coma Scale (GCS), Modified Barthel Index (MBI), modified Ranson Scale (mRS), National Institute of Health Stroke Scale (NIHSS), Jebsen Hand Function Test (JHFT), and Motricity Index (MI). The evaluation of motor outcome was conducted by a rehabilitation department and a resident who was trained professionally in evaluating the function of recovering stroke patients. All of them were not related to the study analysis, they were in charge of evaluation only and were blinded to DTI data.

### 2.3. Diffusion Tensor Tractography

Magnetic resonance imaging examinations were performed using a 3.0 Tesla whole-body scanner (Signa Exite HD, General Electric, Waukesha, WI, USA) with an eight-channel head coil. DTI was performed using the single-shot, spin-echo, echo-planar imaging technique, employing Stejskal–Tanner’s diffusion-sensitizing pulses. The DTI imaging parameters were as follows: 210 × 210 mm^2^ field-of-view, 128 × 128 mm^2^, matrix size (interpolated to 256 × 256), 4 mm slice thickness, 0.8203 × 0.8203 × 4 mm^2^ voxel size, 34 axial slices, 10,000 ms repetition time, 87.2 ms echo time, and b-value of 1000 s/mm^2^. Diffusion was measured in 25 non-collinear directions.

### 2.4. Voxel-Wise Analysis of DTI Metrics

Diffusion tensors were calculated for every voxel using MedINRIA software (Asclepios Research Project; INRIA Sophia Antipolis, Cedex, France, http://www.sop.inria.fr/aasclepios/software/MedINRIA/), which performs the estimation with least squares on the linearized version of the Stejskal and Tanner diffusion equation. In the MedINRIA program, the stepwise analysis method was as follows: (1) Create an FA color map at each time point (pre, post1, post2, post3) in the individual data. (2) On each FA color map, set regions of interest (ROIs) in the blur portion of upper and lower pons to make the corticospinal tract (CST). (3) Extract the fiber tract passing through these two pixels and color the fibers to distinguish which side (left and right hemisphere) they are on in the brain. (4) Extract FA value for each fiber tract. (5) Calculate the average value of the FA value of the fiber tract at each time point by group and separate the tract from the affected and non-affected sides of the brain to identify statistical differences in the mean FA of each group. Fractional anisotropy (FA), a measure of the degree of the directional preference of water diffusion, was calculated for each brain voxel [28].

Tractography was then applied to the DTI data to reconstruct the white matter tracts by successively following the path of the preferred direction of water diffusion. Trilinear Log-Euclidean interpolation was used throughout tracking to produce fibers with smoother trajectories. The fiber tractography parameters were defined as follows: FA threshold, 0.2; sampling pixel number, 1; minimum fiber length, 5 mm; and smoothing of the interpolated fiber, 20%. The other parameters for fiber tracking used the MedINRIA default values [29].

Regions of interest (ROI) were drawn on each patient’s color-coded FA map in the blue portion of the upper (red) and lower (green) pons on the axial image where neuroanatomical and tractography studies located the CST. Regions of interest were given at the anterior blue portion of the pontine basis on the color map at two levels; upper pons—the first axial image on which the superior cerebellar peduncle could be seen and the lower pons—the axial image on which the middle cerebellar peduncle between the mid-pons and pontomedullary junction could be seen [30]. FA values were measured by selecting the areas where the CST tract fibers passed. We compared the FA values at 1 day after stroke onset (pre) and 3 weeks (post1), 3 months (post2), and 6 months (post3) after the initial treatment between the good and poor outcome groups. Currently, the most widely used invariant measure of anisotropy is fractional anisotropy (FA) described originally by Basser and Pierpaoli [31]. We also calculated FA ratio (FA value of the ipsilateral side/FA value of the contralateral side) and compared it between the groups [32]. The evaluation of imaging outcomes was independently performed by one radiologist and one rehabilitation physician with blindness. If there was discrepancy, it was evaluated again and averaged to calculate the final value.

### 2.5. Statistical Analysis

Statistical analysis was performed using SPSS 25.0 for Windows software (IBM Corp., Armonk, NY, USA). The two-sample t-test was used to compare FA values of the affected and non-affected hemispheres between the good and poor outcome groups at pre, post1, post2, and post3. Correlation between FA values and clinical data (GCS, MBI, mRS, NIHSS, JHFT, and MI) in the affected and non-affected hemispheres at each time point was determined using Pearson’s correlation test. *p* < 0.05 was considered significant.

## 3. Results

A total of 12 patients were included in the study (aged 50 ± 12 years). Eight patients were male (aged 46 ± 11 years) and 4 were female (aged 59 ± 9 years).

### 3.1. Patient Characteristics and Diffusion Tensor Tractography (DTT) Data

Patient characteristics of the good and poor outcome groups are summarized in Table 1 and Table 2, respectively. Figure 1 and Figure 2 present the DTT data of the two groups, respectively. There were no significant differences in age (*p* = 0.349, Mann–Whitney U) or sex ratio (*p* = 1.000, Pearson’s chi-square) between the good and poor outcome groups. All imaged CST lines were consistent with anatomic knowledge.

### 3.2. FA Differences in the Affected and Non-Affected Sides of the Brain

Figure 3 shows the mean FA values in the affected and non-affected sides of the brain of the good and poor outcome groups at each time point. The Wilcoxon’s signed-rank test was used to compare values at each time point. In the affected side, the mean FA value was significantly higher in the good outcome group than that in the poor outcome group at pre (*p* = 0.004) and post3 (*p* = 0.025); no significant difference was found between the groups at post1 and post2. No significant difference in mean FA value was found in the non-affected side between the good and poor outcome groups at all time points (* *p* > 0.05).

Figure 4 shows the differences in FA ratio in the affected and non-affected sides of the brain between the good and poor outcome groups at each time point. The FA ratio of the good outcome group was significantly higher than that of the poor outcome group at pre (** *p* < 0.01) and post3 (* *p* < 0.05), but there were no significant differences between the two groups at post1 and post2.

### 3.3. Correlation of FA Values and FA Ratio at Pre with the Sum of BMS Score at Post3

Pearson’s correlation analysis showed that FA values (R = 0.8, ** *p* <0.01) and FA ratio (R = 0.649, * *p* < 0.05) at pre significantly correlated with the sum of BMS scores at post3 (Figure 5).

## 4. Discussion

Retrograde Wallerian degeneration of the CST is related to stroke prognosis [20] and can be evaluated by DTI [11]. Our results demonstrated that initial rFA can predict motor outcome at 6 months in patients with putaminal hemorrhage. The initial rFA positively correlated with motor recovery after 3 months, indicating that initial rFA has the potential to predict motor outcome.

Growing evidence suggests that combining clinical scores with information about corticospinal tract (CST) integrity can improve predictions about motor outcome. The extent of CST damage on DTI and/or the overlap between the CST and a lesion are a key prognostic factor that determines motor performance and outcome [11,12]. Many studies have been conducted to predict motor recovery through DTI parameters in ICH patients [21,22,23,24,25,26]. Kusano et al. demonstrated that relative fractional anisotropy (rFA) can reflect prognosis [21]. They reported that the FA value of the affected side was 11% lower than that of the unaffected side on the second day after stroke onset. This study supports our findings in that rFA reflects motor outcomes. Kuzu et al. showed the FA values of the cerebral peduncle on the pathological side in patients with ICH on day 3 can predict motor function on day 90 [22]. Wang et al. showed that the use of DTI during the early stages of ICH may predict motor outcomes at 6 months after ICH. Moreover, as compared to use of DTI within 3 days of ICH onset, the application of DTI at 2 weeks after ICH could more accurately predict the motor outcomes and daily living activities of patients [23]. Our study is in line with the previous study, in that, the FA value at the time of onset can predict a long-term outcome at 6 months. Koyama et al. developed a model in which FA values assessed by DTI at 2 weeks correlated well with motor outcome at 1 month after onset [26]. However, most of these studies were cross-sectional in design, and a longitudinal change of rFA after ICH and measured motor outcome was previously demonstrated by two studies [24,27]. 

Wang et al. demonstrated a longitudinal change of rFA after ICH and measured motor outcome [27]. They performed DTI 3 days and 2 weeks after stroke onset in 36 ICH patients and compared it with a motor outcome at 6 months. FA, rFA, and MD at the anterior cerebral peduncle level were measured as DTI parameters. The results showed that the rFA at 2 weeks significantly correlated with the mRS at 6 months. The researchers demonstrated that hematoma size itself was not related to motor outcome after stroke, which was previously described by Kusano et al., Puig et al., and Feys et al. [21,33,34]. Although Ma et al. investigated change in FA in acute and chronic hemorrhagic stroke over a 3 month period, longer follow-up is needed because prognosis in patients with a severely impaired CST is difficult to predict [24]. In the longitudinal study, our study was the only one that measured changes in motor outcome and FA up to 6 months. However, applying the proportional recovery model to predict long-term prognosis is difficult in patients with severe motor weakness. Therefore, long-term follow-up studies are of great clinical significance.

We measured outcome with hand and arm BMS, which reflects CST function and integrity [12], and was developed to evaluate recovery in stroke patients. A previous study used the MMT to evaluate hand function [27]. Although MMT is representative of muscle strength, it is less sensitive in measuring stroke recovery [35]. In this study, both hand BMS and the combined hand, arm, and leg recovery after 3 months showed a positive correlation with FA value and ratio, suggesting that the corticorubrospinal tract or corticotegmental-spinal tract may be involved in addition to the CST [3]. Further studies of these tracts are needed.

In this study, rFA value was suggested as an important parameter in predicting long-term motor outcome. However, a few studies have reported that the plasticity of unaffected hemispheres also plays a role in the stroke recovery process [36,37,38]. Bajaj et al. reported that the unaffected hemisphere more accurately reflected the behavioral conditions than the connectivity patterns in the affected hemisphere [37]. Jang et al. demonstrated the fiber number of the CST in the unaffected hemisphere was increased by the change of the dominant hand in stroke patients [38]. Therefore, the plasticity of the unaffected hemisphere and inter-hemispheric balance are also important in stroke recovery, so further analysis is needed in future studies.

In this study, hematoma volume was not associated with prognosis, and hematoma volume did not differ between the two groups. This finding agrees with previous reports [21,22]. We hypothesize that outcome is more dependent on the hematoma’s effect on the CST rather than on hematoma volume. Therefore, ICH that involves the posterior limb of the internal capsule has a greater effect on long-term motor outcome. In this study, we selected ROIs in the anterior pons. Most previous studies included regions in the cerebral peduncle [21,22,27]. However, using the pons is advantageous, because hemosiderin accumulation is less likely to influence the results, although, anatomically, this is further away from the site of injury.

Our study has several limitations. First, we did not compare data between stroke patients and normal healthy individuals. Second, our sample size was small; however, we were able to achieve statistically significant results. Third, the pontine rFA is relatively distant from the site of hemorrhage to reflect its effect on the CST. Future studies should include numerous ROIs around the hematoma and in the corona radiata, cerebral peduncle, and pons. Finally, we did not compare patients who had surgery with those who did not, so the effect of surgery was not examined. As the number of patients will increase in the future, it is necessary to conduct a study comparing prognosis between patients with and without surgery [25]. DTI can also be useful for monitoring treatment response such as physical therapy and mental practice and white matter remodeling after stroke [39]. The combination of DTI, which evaluates the integrity of CST, and resting state-fMRI, which reflects brain connectivity, will provide more insight into the neuroplasticity mechanism after stroke [40,41].

## 5. Conclusions

The change in FA ratio on diffusion tractography can predict motor recovery after hemorrhagic stroke. DTI can be an excellent parameter for predicting prognosis in patients with hemorrhagic stroke with poor prognosis.

## Figures and Tables

**Figure 1 jcm-09-01304-f001:**
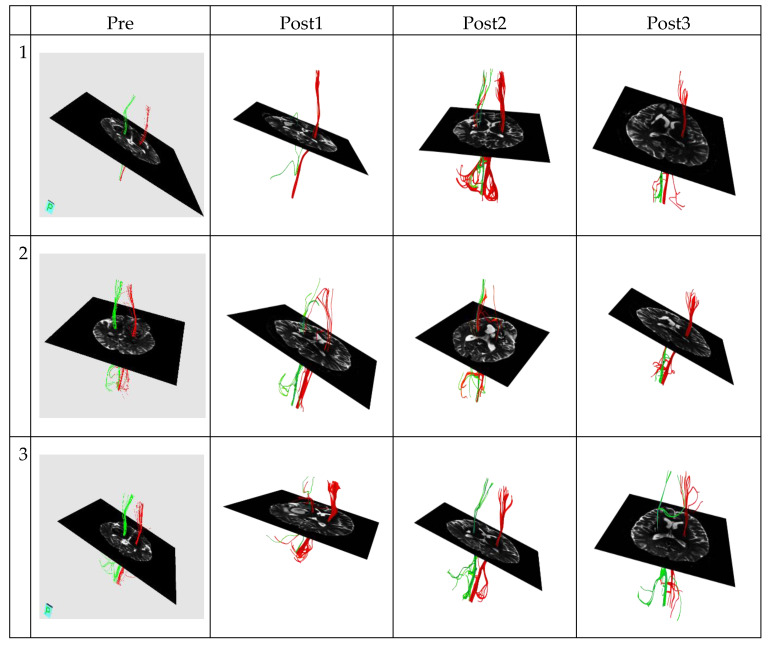
The corticospinal tract diffusion tensor tractography data in the good outcome group of three patients. Green represents the corticospinal tract (CST) on the left hemisphere. Conversely, red represents the corticospinal tract (CST) on the right hemisphere.

**Figure 2 jcm-09-01304-f002:**
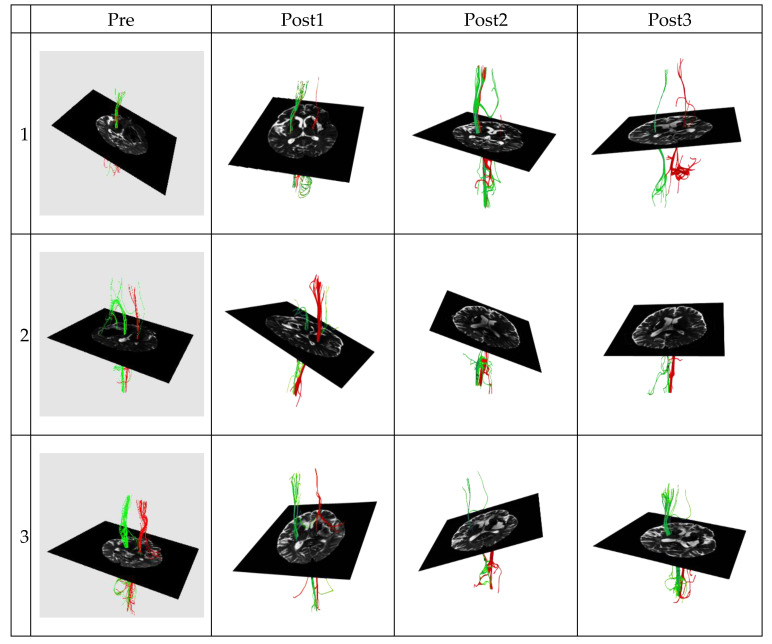
The corticospinal tract diffusion tensor tractography data in the poor outcome group of three patients. Green represents the corticospinal tract (CST) on the left hemisphere. Conversely, red represents the corticospinal tract (CST) on the right hemisphere.

**Figure 3 jcm-09-01304-f003:**
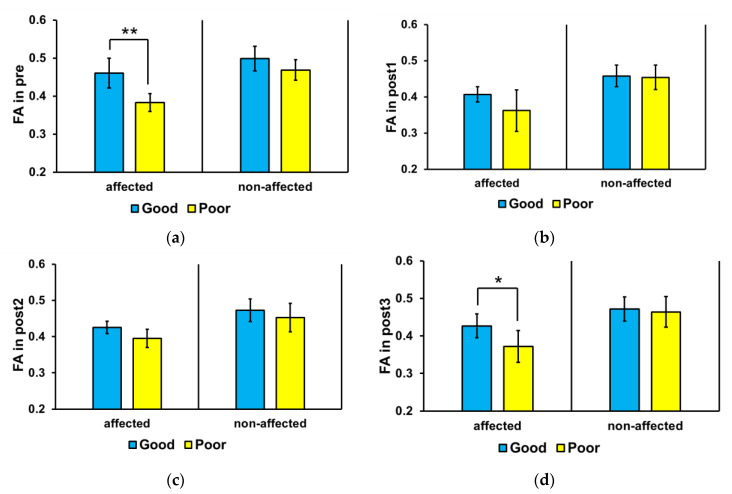
Mean corticospinal tract fractional anisotropy (FA) values in the affected and non-affected sides of the brain. In the affected side, the mean FA value was significantly higher in the good outcome group than that in the poor outcome group at pre (* *p* < 0.01) and post3 (* *p* < 0.05). Abbreviations: (**a**) baseline (pre), (**b**) 3 weeks after stroke onset (post1), (**c**) 3 months (post2), and (**d**) 6 months (post3). ** *p* < 0.01.

**Figure 4 jcm-09-01304-f004:**
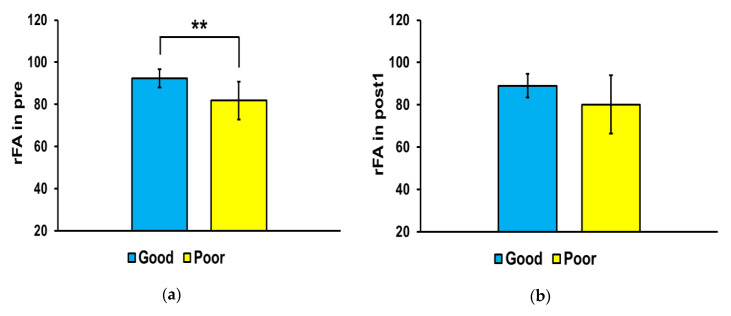
Mean fractional anisotropy (FA) ratios in the corticospinal tract. In the affected side, the mean FA ratio was significantly higher in the good outcome group than that in the poor outcome group at pre (* *p* < 0.01) and post3 (* *p* < 0.05). Abbreviations: (**a**) baseline (pre), (**b**) 3 weeks after stroke onset (post1), (**c**) 3 months (post2), and (**d**) 6 months (post3). ** *p* < 0.01.

**Figure 5 jcm-09-01304-f005:**
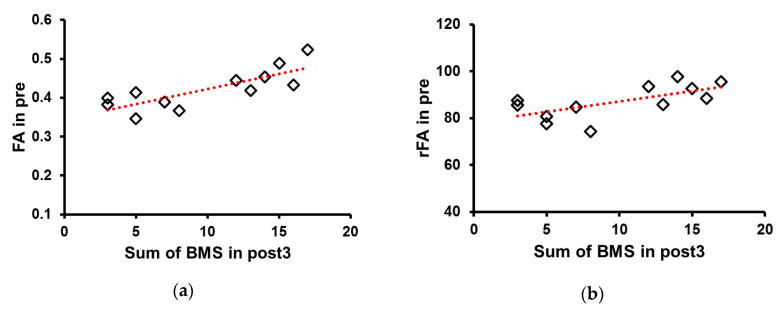
Correlation between fractional anisotropy (FA) value and ratio and the sum of Brunnstrom motor recovery stage (BMS). (**a**) FA values (*R* = 0.8) and (**b**) the FA ratio (*R* = 0.652) at pre significantly correlated with the sum of BMS at post3. Abbreviations: baseline (pre), 3 weeks after stroke onset (post1), 3 months (post2), and 6 months (post3).

**Table 1 jcm-09-01304-t001:** Patient characteristics and diffusion tensor tractography fractional anisotropy values of the good outcome group.

Patient No.	Age (Years)/Sex	Treatment	Location of ICH	Volume of ICH (mL)	Period *	DT Imaging Data		
						FA in Affected Hemisphere	FA in non-Affected Hemisphere	FA Ratio **
1	36/M	Stereotactic aspiration	Lt. Putamen	18.85	Pre	0.52	0.55	95.62
					Post1	0.42	0.44	93.82
					Post2	0.41	0.43	95.52
					Post3	0.41	0.45	92.76
2	57/M	Stereotactic aspiration	Rt. Putamen	35.40	Pre	0.45	0.46	97.72
					Post1	0.43	0.45	96.47
					Post2	0.44	0.48	93.17
					Post3	0.45	0.48	95.25
3	31/M	Stereotactic aspiration	Lt. Basal Ganglia	40.50	Pre	0.49	0.53	92.76
					Post1	0.41	0.47	85.69
					Post2	0.41	0.51	81.46
					Post3	0.47	0.50	92.81
4	48/M	Conservative	Lt. Putamen	11.50	Pre	0.44	0.47	93.57
					Post1	0.41	0.49	85.27
					Post2	0.44	0.50	89.36
					Post3	0.44	0.49	89.17
5	52/F	Stereotactic aspiration	Lt. Basal Ganglia	10.5	Pre	0.42	0.49	85.83
					Post1	0.37	0.41	90.93
					Post2			
					Post3	0.38	0.42	90.53
6	49/F	Conservative	Lt. Basal Ganglia	11.6	Pre	0.43	0.49	88.47
					Post1	0.40	0.49	81.99
					Post2	0.41	0.45	91.74
					Post3	0.41	0.50	82.65

* Baseline (pre) and 3 weeks (post1), 3 months (post2), and 6 months (post3) after stroke onset. ** FA value of the ipsilateral side/FA value of the contralateral side. DT, diffusion tensor; FA, fractional anisotropy; ICH, intracranial hemorrhage.

**Table 2 jcm-09-01304-t002:** Patient characteristics and diffusion tensor tractography fractional anisotropy values of the poor outcome group.

Patient No.	Age (Years)/Sex	Treatment	Location of ICH	Volume of ICH (mL)	Period *	DT Imaging Data		
						FA in Affected Hemisphere	FA in Non-Affected Hemisphere	FA Ratio **
1	46/M	Conservative	Rt. Basal Ganglia	29.061	Pre	0.37	0.49	74.40
					Post1	0.29	0.49	59.11
					Post2	0.39	0.45	87.41
					Post3	0.40	0.46	86.97
2	67/F	Stereotactic aspiration	Rt. Putamen	28.91	Pre	0.38	0.45	85.51
					Post1	0.38	0.44	84.87
					Post2	0.42	0.46	89.97
					Post3	0.30	0.42	71.49
3	69/M	Conservative	Lt. Putamen	27.385	Pre	0.35	0.45	77.71
					Post1	0.35	0.41	86.87
					Post2	0.36	0.40	91.49
					Post3	0.38	0.43	88.12
4	38/M	Conservative	Lt. Basal Ganglia	53.54	Pre	0.41	0.51	80.70
					Post1	0.46	0.47	97.25
					Post2	0.42	0.43	98.05
					Post3	0.38	0.53	70.29
5	70/F	Conservative	Rt. Putamen	18.20	Pre	0.40	0.46	87.64
					Post1	0.33	0.49	68.42
					Post2	0.37	0.47	77.65
					Post3	0.35	0.47	75.25
6	46/M	Conservative	Rt. Basal Ganglia	8.761	Pre	0.39	0.46	84.76
					Post1	0.36	0.43	84.58
					Post2	0.41	0.51	81.56
					Post3	0.43	0.47	90.01

* Baseline (pre) and 3 weeks (post1), 3 months (post2), and 6 months (post3) after stroke onset. ** FA value of the ipsilateral side/FA value of the contralateral side. DT, diffusion tensor; FA, fractional anisotropy; ICH, intracranial hemorrhage.

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
