# Peer review of "Prediction of Motor Recovery in Patients with Basal Ganglia Hemorrhage Using Diffusion Tensor Imaging"

_jcm, 2020, doi:10.3390/jcm9051304_

Round 1

Reviewer 1 Report

Review of the manuscript number jcm-760314

This is an interesting study looking at the viability of using diffusion tensor imaging to predict motor recovery in patients with basal ganglia hemorrhage.

The authors found that pre-treatment fractional anisotropy values using diffusion tensor imaging significantly  correlated with the Brunnstrom recovery stage score at 6 months after treatment. The authors also found higher functional anisotropic values in the group with good clinical outcome after treatment.

The introduction is too brief and there is a lack of background information. Please elaborate on findings of previous studies

Could the authors also please list any exclusion criteria? i.e. any other neurological pathology, history of previous stroke, how many months after the vascular incident etc.

Could the authors please improve the description and add better information about the colors showing  in figures 1 and 2?

The quality of the data presented on figure 3 is poor. Could the authors please improve this figure?

The captions of figures 3, 4 and 5 are too wordy, please simplify and elaborate on the description in the main text.

This study is, in itself interesting and t he results are promising to use this technique to improve outcome prediction in patients who have had putaminal cerebral hemorrhage. However, the presentation of the manuscript is lacking and I would encourage the authors to improve both contents and quality. Please elaborate further in the introduction and conclusions

Also, the authors say that they were able to achieve statistically significant results. I would be careful of this statement given the low number of samples and further use the results to carry out a power calculation to determine whether their sample size is adequate.

Author Response

Response to Reviewer 1 Comments

Thank you for giving me the opportunity to submit a revised draft of my manuscript titled ‘Prediction of Motor Recovery in Patients with Basal Ganglia Hemorrhage using Diffusion Tensor Imaging’ to journal of clinical medicine. We appreciate the time

and effort that you and the reviewers have dedicated to providing your valuable feedback on my manuscript. We are grateful to the reviewers for their insightful comments on my paper. We have been able to incorporate changes to reflect most of the suggestions provided by the reviewers. We have highlighted the changes within the manuscript.

Here is a point-by-point response to the reviewers’ comments and concerns.

This is an interesting study looking at the viability of using diffusion tensor imaging to predict motor recovery in patients with basal ganglia hemorrhage.

The authors found that pre-treatment fractional anisotropy values using diffusion tensor imaging significantly correlated with the Brunnstrom recovery stage score at 6 months after treatment. The authors also found higher functional anisotropic values in the group with good clinical outcome after treatment.

Point 1: The introduction is too brief and there is a lack of background information. Please elaborate on findings of previous studies

Response 1: Thank you for your suggestion. We have revised the introduction section to be more detailed and to provide background information as folllows:

“The most common mechanism of neuronal injury after hemorrhagic stroke is Wallerian degeneration, which primarily occurs in the corticospinal tract (CST) both adjacent and distant to the hematoma site [11]. Growing evidence shows the potential of diffusion tensor imaging (DTI) as a noninvasive magnetic resonance imaging (MRI) technique for in vivo quantification of microstructural damage to white matter (WM) tracts following stroke [12]. Conventional non-invasive imaging with conventional MRI is not sensitive enough to obtain information about extent of damage to guide management strategies more effectively [11]. Essentially the main parameter that is measured in DTI is the degree of fractional anisotropy (FA) in a given voxel of the precessing proton and its Eigen vector in an ellipsoid domain[11]. Recently, new diffusion measures—quantitative anisotropy (QA) introduced to the field of DTI for the analysis of diffusion properties of white matter [13,14]. QA represent how much water diffuses (i.e., density) in a specific/restricted direction and in an isotropic fashion (i.e., total isotropic component), respectively. In addition, the difference between QA and FA pertains to the fact that QA is a measure of water diffusion along each fiber orientation, whereas FA is defined for each voxel. Compared to FA, QA is also reported to have lower susceptibility to partial volume effects of crossing fibers, free-water diffusion in ventricles, and non-diffusive particles [14]. Despite the advantages of QA, FA is the most basic parameter and the degree of white matter integrity can be intuitively compared by comparing the ratio of bilateral hemispheres, and it can be compared with the previous studies which using FA value.

Byblow et al. showed the value of DTI in the subacute phase in identifying which initially severe patients have some potential for recovery and which do not. Other predictors usually have only a moderate predictive effect such as infarct volume, age, gender, dose of rehabilitation treatments, comorbidities, and stroke subtype [2,15–18]. DTI is increasingly demonstrating greater accuracy in detecting microstructural damage related to Wallerian degeneration after acute stroke [19,20].          

Several studies have investigated its utility in predicting prognosis after intracranial hemorrhage (ICH)[21–26]. Kusano et al. demonstrated that relative fractional anisotropy (rFA) can reflect prognosis [21]. Other researchers have also shown similar results. Kuzu et al showed the FA values ​​of the cerebral peduncle on the pathological side in patients with ICH on day 3 can predict on-day 90 motor function [22]. Wang et al showed that the use of DTI during the early stages of ICH may predict motor outcomes at 6 months after ICH. Moreover, as compared to use of DTI within 3 days of ICH onset, the application of DTI at 2 weeks after ICH could more accurately predict the motor outcomes and daily living activities of patients [23]. Koyama et al developed a model which FA values assessed by DTI at 2 weeks correlated well with motor outcome at 1 month after onset [26]. However, most of these studies were cross sectional in design, and only two provided a longitudinal evaluation [24,27]. Wang et al. demonstrated a longitudinal change of rFA after ICH and measured motor outcome with manual muscle testing (MMT), which is too broad to reflect the integrity of the CST [27]. In addition, patients were followed up for only 2 weeks after onset in their study. Although Ma et al. investigated change in FA in acute and chronic hemorrhagic stroke over a 3-month period, longer follow-up is needed because prognosis in patients with a severely impaired CST is difficult to predict [24].

Therefore, this study aimed to investigate the value of diffusion tensor imaging in predicting motor recovery after basal ganglia hemorrhage. The hypothesis of the study is that the initial FA value of DTI imaging after stroke onset would reflect the motor outcome of recovery.“

Point 2: Could the authors also please list any exclusion criteria? i.e. any other neurological pathology, history of previous stroke, how many months after the vascular incident etc.

Response 2: Thank you for your kind correction. We revised the exclusion criteria in more detail as you recommended as follows:

“The exclusion criteria are as follows. 1) history of previous stroke 2) any other neurological 3) unstable medical condition pathology 4) follow-up loss.”

Point 3: Could the authors please improve the description and add better information about the colors showing  in figures 1 and 2?

Response 3:  Thank you for kind comments. We revised figure legends as follows:

“Figure 1. The corticospinal tract diffusion tensor tractography data in the good outcome group of three patients. The green in pre and purple color in post1, 2, and 3 represent the corticospinal tract (CST) on the left hemisphere. Conversely, the red in pre and orange color in post1, 2, and 3 represent the corticospinal tract (CST) on the right hemisphere”

“Figure 2. The corticospinal tract diffusion tensor tractography data in the poor outcome group of three patients. The green in pre and purple color in post1, 2, and 3 represent the corticospinal tract (CST) on the left hemisphere. Conversely, the red in pre and orange color in post1, 2, and 3 represent the corticospinal tract (CST) on the right hemisphere.”

Point 4: The quality of the data presented on figure 3 is poor. Could the authors please improve this figure?

Response 4:  Thanks for the kind comments. As you pointed out, figure 3 has been saved again for high readability and high resolution.

(a)

(b)

(c)

(d)

Point 5: The captions of figures 3, 4 and 5 are too wordy, please simplify and elaborate on the description in the main text.

Response 5: Thanks for the kind comments. We simplified the captions of figures 3, 4 and 5 and elaborated on the description in the main text as follows:

“Figure 3. Mean corticospinal tract fractional anisotropy (FA) values in the affected and non-affected sides of the brain. In the affected side, the mean FA value was significantly higher in the good outcome group than the poor outcome group at pre (*P < 0.01) and post3 (*P < 0.05).

Figure 4. Mean fractional anisotropy (FA) ratios in the corticospinal tract. In the affected side, the mean FA ratio was significantly higher in the good outcome group than the poor outcome group at pre (*P < 0.01) and post3 (*P < 0.05).

Figure 5. Correlation between fractional anisotropy (FA) value and ratio and the sum of BMS. (a) FA values (R = 0.8, **P <0.01) and (b) the FA ratio (R = 0.652, *P < 0.05) at pre significantly correlated with the sum of BMS at post3. “

Point 6: This study is, in itself interesting and t he results are promising to use this technique to improve outcome prediction in patients who have had putaminal cerebral hemorrhage. However, the presentation of the manuscript is lacking and I would encourage the authors to improve both contents and quality. Please elaborate further in the introduction and conclusions

Response 6: Thank you for your suggestion.

We elaborated further in the introduction and conclusions as follows:

Introduction

“Prediction of prognosis in patients with hemorrhagic stroke is difficult but important when planning and designing an individual patient’s rehabilitation strategy [1]. Many studies have used clinical features and neurophysiological findings to predict prognosis, such as the Predict Recovery Potential (PREP) model, which is based on the proportional recovery theory [2,3]. However, most of these studies investigated patients with ischemic stroke rather than hemorrhagic stroke [4,5]. The pathogenesis and recovery mechanisms of hemorrhagic and ischemic stroke are different [6–10]; therefore, models developed for ischemic stroke do not necessarily apply to hemorrhagic stroke patients [11].

The most common mechanism of neuronal injury after hemorrhagic stroke is Wallerian degeneration, which primarily occurs in the corticospinal tract (CST) both adjacent and distant to the hematoma site [11]. Growing evidence shows the potential of diffusion tensor imaging (DTI) as a noninvasive magnetic resonance imaging (MRI) technique for in vivo quantification of microstructural damage to white matter (WM) tracts following stroke [12]. Conventional non-invasive imaging with conventional MRI is not sensitive enough to obtain information about extent of damage to guide management strategies more effectively [11]. Essentially the main parameter that is measured in DTI is the degree of fractional anisotropy (FA) in a given voxel of the precessing proton and its Eigen vector in an ellipsoid domain[11]. Recently, new diffusion measures—quantitative anisotropy (QA) introduced to the field of DTI for the analysis of diffusion properties of white matter [13,14]. QA represent how much water diffuses (i.e., density) in a specific/restricted direction and in an isotropic fashion (i.e., total isotropic component), respectively. In addition, the difference between QA and FA pertains to the fact that QA is a measure of water diffusion along each fiber orientation, whereas FA is defined for each voxel. Compared to FA, QA is also reported to have lower susceptibility to partial volume effects of crossing fibers, free-water diffusion in ventricles, and non-diffusive particles [14]. Despite the advantages of QA, FA is the most basic parameter and the degree of white matter integrity can be intuitively compared by comparing the ratio of bilateral hemispheres, and it can be compared with the previous studies which using FA value.

Byblow et al. showed the value of DTI in the subacute phase in identifying which initially severe patients have some potential for recovery and which do not. Other predictors usually have only a moderate predictive effect such as infarct volume, age, gender, dose of rehabilitation treatments, comorbidities, and stroke subtype [2,15–18]. DTI is increasingly demonstrating greater accuracy in detecting microstructural damage related to Wallerian degeneration after acute stroke [19,20].          

Several studies have investigated its utility in predicting prognosis after intracranial hemorrhage (ICH)[21–26]. Kusano et al. demonstrated that relative fractional anisotropy (rFA) can reflect prognosis [21]. Other researchers have also shown similar results. Kuzu et al showed the FA values ​​of the cerebral peduncle on the pathological side in patients with ICH on day 3 can predict on-day 90 motor function [22]. Wang et al showed that the use of DTI during the early stages of ICH may predict motor outcomes at 6 months after ICH. Moreover, as compared to use of DTI within 3 days of ICH onset, the application of DTI at 2 weeks after ICH could more accurately predict the motor outcomes and daily living activities of patients [23]. Koyama et al developed a model which FA values assessed by DTI at 2 weeks correlated well with motor outcome at 1 month after onset [26]. However, most of these studies were cross sectional in design, and only two provided a longitudinal evaluation [24,27]. Wang et al. demonstrated a longitudinal change of rFA after ICH and measured motor outcome with manual muscle testing (MMT), which is too broad to reflect the integrity of the CST [27]. In addition, patients were followed up for only 2 weeks after onset in their study. Although Ma et al. investigated change in FA in acute and chronic hemorrhagic stroke over a 3-month period, longer follow-up is needed because prognosis in patients with a severely impaired CST is difficult to predict [24].

Therefore, this study aimed to investigate the value of diffusion tensor imaging in predicting motor recovery after basal ganglia hemorrhage. The hypothesis of the study is that the initial FA value of DTI imaging after stroke onset would reflect the motor outcome of recovery.”

Conclusion

“The change in FA ratio on diffusion tractography can predict motor recovery after hemorrhagic stroke. DTI can be an excellent parameter for predicting prognosis in patients with hemorrhagic stroke with poor prognosis.”

Point 7: Also, the authors say that they were able to achieve statistically significant results. I would be careful of this statement given the low number of samples and further use the results to carry out a power calculation to determine whether their sample size is adequate.

Response 7: Thank you for your thoughtful comments. I strongly agree with your opinion. Due to the small sample size, it is difficult to generalize our conclusions. This study is a priliminary and exploratory study, and we will carry out a follow-up study to confirm this by increasing the number of subjects in the future. As you pointed out, it is possible that a very significant correlation result was observed due to the small number of subjects. It is possible that the results of this statistical verification could be swallowed if the number of subjects increases through future expansion studies. More number of patients will be needed to confirm this correlation.

Reviewer 2 Report

Summary

The authors longitudinally investigated the measures of fractional anisotropy (FA) and motor recovery in individuals with basal ganglia hemorrhage stroke. Authors used diffusion tensor imaging (DTI) technique on a sample of 12 participants. ‘Longitudinal’ nature is the strength and the ‘small sample size’ is the weakness of the study. In addition, I have following questions and concerns regarding this manuscript:

Abstract

  • I would suggest rephrasing the term “value of diffusion tensor imaging” in abstract and throughout the manuscript.
  • Please include mean age and SD of men and women separately as well (along with over all mean age and SD).
  • Authors reported greater FA in the affected side of good outcome group, but did not specify the regions of the affected side.

Introduction

  • Lines 48-49: Authors mentioned “several studies” but cited only one study.
  • Lines 50-51: Authors mentioned that most of the studies investigated patients with ischemic stroke but did not cite any study.
  • Please provide references for claims made in lines 51-53.
  • Lines 56-57: DTI is not a recent approach for white matter integrity, people are using this from several years now. Researchers are now opting for white matter compactness (quantitative anisotropy - QA) over white matter integrity (FA). A brief discussion over literature using QA vs. FA and its utilization (e.g., https://www.frontiersin.org/articles/10.3389/fneur.2017.00616/full) and authors’ preference of using FA over QA would be beneficial for the readers.
  • Why did the authors prefer to use FA only? Multiple DTI measures are always recommended to be calculated for DTI studies, e.g. MD, RD, FA, QA, fiber density, tract volume etc. A general introduction of these parameters and about DTI in general is missing.
  • Why did authors prefer DTI technique over fMRI and brain morphometry? A clear rationale of the study is missing.
  • Previous literature as described above is not cited and discussed in detail.
  • Please describe a clear Hypothesis and aims of the manuscript.

Methods

  • Its mentioned that participants were aged 20 and above, and mean age is 52. Please provide overall sample age range, and gender specific age details.
  • Were there any sex differences in age and other demographics? Were there any covariates included?
  • What was the voxel size used for DTI data?
  • Sample size for the study is really small. For DTI studies, how did the authors make sure whether the sample size is sufficiently enough?
  • What was criteria and method used to draw ROIs? Calculations about estimating FA for these ROIs is not clear.
  • Step-wise analysis details performed in MedINRIA would be helpful for readers and for replication purposes.

Results

  • Table 1 of good outcome group has 6 participants data, but Figure 1 has fiber tracts of only 3 participants. Similarly, Table 2 of poor outcome group has 6 participants data, but Figure 2 has fiber tracts of only 3 participants. Please clarify this in figure legends.
  • Please clarify the meaning of different colors of fiber tracts. Also, please provide color bars of FA magnitude.
  • Figure legends of figures 3 and 4 are not clear. For instance, figure labels, (a), (b) etc. are not described in the legend. Also, I would prefer these figures to be color coded for affected and non-affected hemispheres.
  • Significant differences described by * in figure 3a and 3d are not clear.
  • In figure 5, it’s not clear why data points (i.e., 10) are less than the number of subjects (i.e., 12).
  • It’s not clear why did the authors correlated FA measures at ‘pre’ with BMS measures at post3 i.e., why between two different conditions.
  • It seems the correlations are highly significant because of smaller size. How would authors justify the strong correlations in figure 5?

Discussion

  • Authors focused their findings on affected hemisphere. However, unaffected hemisphere has been shown to play a significant role in stroke survivors. See: https://www.frontiersin.org/articles/10.3389/fnhum.2016.00650/full. A brief discussion on these previous findings and its significance related to the findings reported by the authors would be beneficial.
  • I am not sure why authors avoided detailed discussion of the results, impact and significance of findings, significance of FA over other parameters, generalization of DTI findings for fMRI and morphometry, and future directions/goals etc.
  • Any detailed discussion on prior work on stroke and stroke therapies such as physical therapy, mental practice and brain connectivity is missing throughout.

Author Response

Response to Reviewer 2 Comments

We thank the reviewer for his/her attention and thoughtful comments.

Thank you for giving me the opportunity to submit a revised draft of my manuscript titled ‘Prediction of Motor Recovery in Patients with Basal Ganglia Hemorrhage using Diffusion Tensor Imaging’ to journal of clinical medicine. We appreciate the time

and effort that you and the reviewers have dedicated to providing your valuable feedback on my manuscript. We are grateful to the reviewers for their insightful comments on my paper. We have been able to incorporate changes to reflect most of the suggestions provided by the reviewers. We have highlighted the changes within the manuscript.

Here is a point-by-point response to the reviewers’ comments and concerns.

Summary

The authors longitudinally investigated the measures of fractional anisotropy (FA) and motor recovery in individuals with basal ganglia hemorrhage stroke. Authors used diffusion tensor imaging (DTI) technique on a sample of 12 participants. ‘Longitudinal’ nature is the strength and the ‘small sample size’ is the weakness of the study. In addition, I have following questions and concerns regarding this manuscript:

Abstract

Point 1:  I would suggest rephrasing the term “value of diffusion tensor imaging” in abstract and throughout the manuscript.

Response 1: We thank the reviewer for kind correction. We rephrased the term “value of diffusion tensor imaging” in abstract and throughout the manuscript. The revised part is marked as hightlight in the manuscript.

Point 2:  Please include mean age and SD of men and women separately as well (along with over all mean age and SD).

Response 2: We thank the reviewer for kind comment. Following the comment by the reviewer, we revised the mean include mean age and SD of men and women separately as well along with over all mean age and SD as follows:

“A total of 12 patients with putaminal hemorrhage were included in the study (aged 50±12 years), eight patients were male (aged 46±11 years) and 4 were female (aged 59 ± 9 years).”

Point 3:  Authors reported greater FA in the affected side of good outcome group, but did not specify the regions of the affected side.

Response 3: We thank the reviewer for this thoughtful and important comment.

We specify the regions of the affected side as follows:

“In the affected side of the brain, mean fractional anisotropy (FA) value on pons was significantly higher in the good outcome group than the poor outcome group at pre (P = 0.004) and post3 (P = 0.025).”

Introduction

Point 4:  Lines 48-49: Authors mentioned “several studies” but cited only one study.

Response 4: We thank the reviewer for very kind comment. We added references as follows:

“Several studies have investigated its utility in predicting prognosis after intracranial hemorrhage (ICH) [21–26].”

Point 5:  Lines 50-51: Authors mentioned that most of the studies investigated patients with ischemic stroke but did not cite any study.

Response 5: We thank the reviewer for very kind comment. We added references as follows:

“However, most of these studies investigated patients with ischemic stroke rather than hemorrhagic stroke [4,5].”

Point 6:  Please provide references for claims made in lines 51-53.

Response 6: Thank you for your kind comment, we added references as follows:

“Many studies have used clinical features and neurophysiological findings to predict prognosis, such as the Predict Recovery Potential (PREP) model, which is based on the proportional recovery theory [2,3]. However, most of these studies investigated patients with ischemic stroke rather than hemorrhagic stroke [4,5]. The pathogenesis and recovery mechanisms of hemorrhagic and ischemic stroke are different [6–10]; therefore, models developed for ischemic stroke do not necessarily apply to hemorrhagic stroke patients [11].”

Point 7:  Lines 56-57: DTI is not a recent approach for white matter integrity, people are using this from several years now. Researchers are now opting for white matter compactness (quantitative anisotropy - QA) over white matter integrity (FA). A brief discussion over literature using QA vs. FA and its utilization (e.g., https://www.frontiersin.org/articles/10.3389/fneur.2017.00616/full) and authors’ preference of using FA over QA would be beneficial for the readers.

Response 7: We thank the reviewer for this very important and thoughtful comment. We completely agree with the reviewer on that it should be discussed white matter compactness (quantitative anisotropy - QA) over white matter integrity (FA). Therefore, we added it as follows:

“Recently, new diffusion measures—quantitative anisotropy (QA) introduced to the field of DTI for the analysis of diffusion properties of white matter [13,14]. QA represent how much water diffuses (i.e., density) in a specific/restricted direction and in an isotropic fashion (i.e., total isotropic component), respectively. In addition, the difference between QA and FA pertains to the fact that QA is a measure of water diffusion along each fiber orientation, whereas FA is defined for each voxel. Compared to FA, QA is also reported to have lower susceptibility to partial volume effects of crossing fibers, free-water diffusion in ventricles, and non-diffusive particles [14]. Despite the advantages of QA, FA is the most basic parameter and the degree of white matter integrity can be intuitively compared by comparing the ratio of bilateral hemispheres, and it can be compared with the previous studies which using FA value.”

Point 8: Why did authors prefer DTI technique over fMRI and brain morphometry? A clear rationale of the study is missing.

Response 8: We thank the reviewer for very thoughtful comment. We completely agree with the reviewer. We added clear rationale of the study as follows:

“The most common mechanism of neuronal injury after hemorrhagic stroke is Wallerian degeneration, which primarily occurs in the corticospinal tract (CST) both adjacent and distant to the hematoma site [11]. Growing evidence shows the potential of diffusion tensor imaging (DTI) as a noninvasive magnetic resonance imaging (MRI) technique for in vivo quantification of microstructural damage to white matter (WM) tracts following stroke [12]. Conventional non-invasive imaging with conventional MRI is not sensitive enough to obtain information about extent of damage to guide management strategies more effectively [11]. Essentially the main parameter that is measured in DTI is the degree of fractional anisotropy (FA) in a given voxel of the precessing proton and its Eigen vector in an ellipsoid domain [11].”

Point 9: Previous literature as described above is not cited and discussed in detail.

Response 9: Thank you for your kind correction. We added citation and discussion in detail as follows:

“Several studies have investigated its utility in predicting prognosis after intracranial hemorrhage (ICH) [21–26]. Kusano et al. demonstrated that relative fractional anisotropy (rFA) can reflect prognosis [21]. Other researchers have also shown similar results. Kuzu et al showed the FA values of the cerebral peduncle on the pathological side in patients with ICH on day 3 can predict on-day 90 motor function [22]. Wang et al showed that the use of DTI during the early stages of ICH may predict motor outcomes at 6 months after ICH. Moreover, as compared to use of DTI within 3 days of ICH onset, the application of DTI at 2 weeks after ICH could more accurately predict the motor outcomes and daily living activities of patients [23]. Koyama et al developed a model which FA values assessed by DTI at 2 weeks correlated well with motor outcome at 1 month after onset [26]. However, most of these studies were cross sectional in design, and only two provided a longitudinal evaluation [24,27]. Wang et al. demonstrated a longitudinal change of rFA after ICH and measured motor outcome with manual muscle testing (MMT), which is too broad to reflect the integrity of the CST [27]. In addition, patients were followed up for only 2 weeks after onset in their study. Although Ma et al. investigated change in FA in acute and chronic hemorrhagic stroke over a 3-month period, longer follow-up is needed because prognosis in patients with a severely impaired CST is difficult to predict [24].”

Point 10:  Please describe a clear Hypothesis and aims of the manuscript.

Response 10: Thank you for your kind comments. We added a clear hypothesis and aims of the manuscript as follows: 

“Therefore, this study aimed to investigate the value of diffusion tensor imaging in predicting motor recovery after basal ganglia hemorrhage. The hypothesis of the study is that the initial FA value of DTI imaging after stroke onset would reflect the motor outcome of recovery.”

Methods

Point 11:  Its mentioned that participants were aged 20 and above, and mean age is 52. Please provide overall sample age range, and gender specific age details.

Response 11: Thank you for your kind comments. We revised overall sample age range, and gender specific age details as follows:

“A total of 12 patients were included in the study (aged 50±12 years). Eight patients were male (aged 46±11 years) and 4 were female (aged 59 ± 9 years).”

Point 12:  Were there any sex differences in age and other demographics? Were there any covariates included?

Response 12: Thank you for your kind comments. There was no difference in age and gender distribution between the two groups, and this was statistically verified.

  1. There was no difference in age distribution between the two groups with p = 0.349 in the mann-whitney U test (P = 0.349).
  2. There was no difference in gender distribution between the two groups in the chi-square test (P = 1.000).

Therefore, the following description was added in the manuscript.

“There were no significant differences in age [P = 0.349, mann-whitney U] or sex [P = 1.000, Pearson’s Chi- square] between the good and poor outcome groups.”

Point 13:  What was the voxel size used for DTI data?

Response 13:

Thank you for your thoughtful comments. In the method section, it was modified as follows:

" The DTI imaging parameters were as follows: 210 × 210  field-of-view, 128 × 128 , matrix size (interpolated to 256 x 256), 4 mm slice thickness, 0.8203 x 0.8203 x 4  voxel size, 34 axial slices, 10000 ms repetition time, 87.2 ms echo time, and b-value of 1000 s/mm2. Diffusion was measured in 25 non-collinear directions.”

Point 14:  Sample size for the study is really small. For DTI studies, how did the authors make sure whether the sample size is sufficiently enough?

Response 14: Thank you for your thoughtful comments. Due to the small sample size, it is difficult to generalize our conclusions. This study is a preliminary and exploratory study, and we will carry out a follow-up study to confirm this by increasing the number of subjects in the future.

Point 15: What was criteria and method used to draw ROIs? Calculations about estimating FA for these ROIs is not clear.

Response 15:  Thank you for your thoughtful comments.  We revised criteria and method used to draw ROIs and calculations about estimating FA for these ROIs as follows:

“Regions of interest (ROI) were drawn on each patient’s color-coded FA map in the blue portion of the upper (red color) and lower (green color) pons on the axial image where neuroanatomical and tractography studies located the CST. Regions of interest were given at the anterior blue portion of the pontine basis on the color map at three levels; upper pons-the first axial image on which the superior cerebellar peduncle could be seen and the lower pons- the axial image on which the middle cerebellar peduncle between the mid-pons and pontomedullary junction could be seen [30].”

“Currently, the most widely used invariant measure of anisotropy is the fractional

anisotropy (FA) described originally by Basser and Pierpaoli [30]”

Point 16:  Step-wise analysis details performed in MedINRIA would be helpful for readers and for replication purposes.

Response 16:

Thank you for your thoughtful comments. We revised the manuscript as follows :

. In the MedINRIA program, the step-wise analysis method was as follows: 1) Create a FA color map at each time point (pre, post1, post2, post3) in the individual data. 2) On each FA color map, set ROIs in the blur portion of upper and lower pons to make corticospinal tract (CST). 3) Extract the fiber tract passing through these two pixels and the fibers are colored to distinguish which side (left and right hemisphere) they are in the brain. 4) Extract FA value for each fiber tract. 5) Calculate the average value of the FA value of the fiber tract at each time point by group and separate the tract from the affected and non-affected sides of the brain to identify statistical differences in the mean FA of each group.

Results

Point 17:  Table 1 of good outcome group has 6 participants data, but Figure 1 has fiber tracts of only 3 participants. Similarly, Table 2 of poor outcome group has 6 participants data, but Figure 2 has fiber tracts of only 3 participants. Please clarify this in figure legends.

Response 17: Thank you for your thoughtful comments. We revised the figure legends more clearly as follows :

“Figure 1. The corticospinal tract diffusion tensor tractography data in the good outcome group of three patients. The green in pre and purple color in post1, 2, and 3 represent the corticospinal tract (CST) on the left hemisphere. Conversely, the red in pre and orange color in post1, 2, and 3 represent the corticospinal tract (CST) on the right hemisphere”

“Figure 2. The corticospinal tract diffusion tensor tractography data in the poor outcome group of three patients. The green in pre and purple color in post1, 2, and 3 represent the corticospinal tract (CST) on the left hemisphere. Conversely, the red in pre and orange color in post1, 2, and 3 represent the corticospinal tract (CST) on the right hemisphere.”

Point 18:  Please clarify the meaning of different colors of fiber tracts. Also, please provide color bars of FA magnitude.

Response 18:  Thank you for your thoughtful comments and suggestion.

The color of the fiber tract is simply to distinguish the two sides (left and right hemisphere), The intensity of the color did not represent the volume of the tract. If the volume of the tract is also studied in a further analysis study, I will put a color bar to display it.

Point 19:  Figure legends of figures 3 and 4 are not clear. For instance, figure labels, (a), (b) etc. are not described in the legend. Also, I would prefer these figures to be color coded for affected and non-affected hemispheres.

Response 19: Thank you for your thoughtful comments. We revised the figure legend to be more clearly by marking the figure label and color coded it on the graph as follows:

(a)

(b)

(c)

(d)

Figure 3. Mean corticospinal tract fractional anisotropy (FA) values in the affected and non-affected sides of the brain. In the affected side, the mean FA value was significantly higher in the good outcome group than the poor outcome group at pre (*P < 0.01) and post3 (*P < 0.05). Abbreviations: (a) baseline (pre), (b) 3 weeks after stroke onset (post1), (c) 3 months (post2), and (d) 6 months (post3).

(a)

(b)

(c)

(d)

Figure 4. Mean fractional anisotropy (FA) ratios in the corticospinal tract. In the affected side, the mean FA ratio was significantly higher in the good outcome group than the poor outcome group at pre (*P < 0.01) and post3 (*P < 0.05). Abbreviations: (a) baseline (pre), (b) 3 weeks after stroke onset (post1), (c) 3 months (post2), and (d) 6 months (post3).

Point 20:  Significant differences described by * in figure 3a and 3d are not clear.

Response 20:

Thank you for your detailed comments. We modified the figure to clarify the "*" mark as above in figure 3a and 3d.

Point 21:  In figure 5, it’s not clear why data points (i.e., 10) are less than the number of subjects (i.e., 12).

Response 21: Thank you for your detailed comments. We revised the figure as follows so that 12 people's data can be visualized well.

(a)

(b)

Figure 5. Correlation between fractional anisotropy (FA) value and ratio and the sum of BMS. (a) FA values (R = 0.8, **P <0.01) and (b) the FA ratio (R = 0.652, *P < 0.05) at pre significantly correlated with the sum of BMS at post3. Abbreviations: baseline (pre), 3 weeks after stroke onset (post1), 3 months (post2), and 6 months (post3).

Point 22:  It’s not clear why did the authors correlated FA measures at ‘pre’ with BMS measures at post3 i.e., why between two different conditions.

Response 22: Thank you for your thoughtful comments. The purpose of this study was to determine whether the initial FA value of the stroke onset can reflect the long-term outcome during the recovery process. Therefore, we analyzed whether the FA value at pre-point was correlated with motor outcomes after 6 months.

Point 23:  It seems the correlations are highly significant because of smaller size. How would authors justify the strong correlations in figure 5?

Response 23: Thank you for your thoughtful comments. I strongly agree with your opinion. As you pointed out, it is possible that a very significant correlation was observed due to the small number of subjects. It is possible that the results of this statistical verification could be swallowed if the number of subjects increases through future expansion studies. More number of patients will be needed to confirm this correlation.

Discussion

Point 24:  Authors focused their findings on affected hemisphere. However, unaffected hemisphere has been shown to play a significant role in stroke survivors. See: https://www.frontiersin.org/articles/10.3389/fnhum.2016.00650/full. A brief discussion on these previous findings and its significance related to the findings reported by the authors would be beneficial.

Response 24: Thank you for your thoughtful comments.

We added A brief discussion on these previous findings and its significance related to the findings and modified the manuscript as follows:

“In this study, rFA value was suggested as an important parameter in predicting long-term motor outcome. However, a few studies have reported that the plasticity of unaffected hemipheres also plays a role in the stroke recovery process [36–38]. Bajaj et al repoerted unaffected hemisphere more accurately reflected the behavioral conditions than the connectivity patterns in the affected hemisphere [37]. Jang et al demonstrated the fiber number of the CST in the unaffected hemisphere was increased by the change of the dominant hand in stroke patients [38]. Therefore, the plasticity of the unffected hemisphere and inter-hemispheric balance are also important in stroke recovery, so further analysis is needed in future studies.”

Point 25:  I am not sure why authors avoided detailed discussion of the results, impact and significance of findings, significance of FA over other parameters, generalization of DTI findings for fMRI and morphometry, and future directions/goals etc.

Response 25:  Thank you for your thoughtful and detailed suggestion.

We added impact and significance of findings, significance of FA over other parameters, generalization of DTI findings for fMRI and morphometry, and future directions/goals and modified the manuscript as follows:

“Growing evidence suggests that combining clinical scores with information about corticospinal tract (CST) integrity can improve predictions about motor outcome. The extent of CST damage on DTI and/or the overlap between the CST and a lesion are key prognostic factor that determines motor performance and outcome [11,12]. Many studies have been conducted to predict motor recovery through DTI parameters in ICH patients.[21–26]. Kusano et al. demonstrated that relative fractional anisotropy (rFA) can reflect prognosis [21]. They reported that the FA value of the affected side was 11% lower than that of the unaffected side on the second day after stroke onset. This study supports our findings in that rFA reflects motor outcomes. Kuzu et al showed the FA values ​​of the cerebral peduncle on the pathological side in patients with ICH on day 3 can predict on-day 90 motor function[22]. Wang et al showed that the use of DTI during the early stages of ICH may predict motor outcomes at 6 months after ICH. Moreover, as compared to use of DTI within 3 days of ICH onset, the application of DTI at 2 weeks after ICH could more accurately predict the motor outcomes and daily living activities of patients [23]. Our study is in line with the previous study in that the FA value at the time of onset can predict a long-term outcome at 6 months. Koyama et al developed a model which FA values assessed by DTI at 2 weeks correlated well with motor outcome at 1 month after onset[26]. However, most of these studies were cross sectional in design, and a longitudinal change of rFA after ICH and measured motor outcome previously demonstrated by two studies [24,27]. Wang et al. demonstrated a longitudinal change of rFA after ICH and measured motor outcome [27]. They performed DTI 3 days and 2 weeks after stroke onset in 36 ICH patients and compared it with a motor outcome of 6 months. FA, rFA and MD at the anterior cerebral peduncle level were measured as DTI parameters. The results showed that the rFA at 2 weeks significantly correlated with the mRS at 6 months. The researchers demonstrated that hematoma size itself was not related to motor outcome after stroke, which was previously described by Kusano et al, Puig et al and Feys et al. [21,33,34]. Although Ma et al. investigated change in FA in acute and chronic hemorrhagic stroke over a 3-month period, longer follow-up is needed because prognosis in patients with a severely impaired CST is difficult to predict [24]. In the longuitudinal study, our study was the only one that measured changes in motor outcome and FA up to 6 months. However, applying the proportional recovery model to predict long-term prognosis is difficult in patients with severe motor weakness. Therefore, long-term follow-up studies are of great clinical significance.”

“DTI can also be useful for monitoring treatment reponse such as physical therapy and mental practice and white matter remodelling after stroke [39]. The combination of DTI, which evaluates the integrity of CST, and resting state-fMRI, which reflects brain connectivity, will provide more insight into the neuroplasticity mechanism after stroke [40,41].”

Point 26:  Any detailed discussion on prior work on stroke and stroke therapies such as physical therapy, mental practice and brain connectivity is missing throughout.

Response 26: Thank you for your thoughtful and detailed suggestion.

We added the detailed discussion on prior work on stroke and stroke therapies such as physical therapy, mental practice and brain connectivity as follows:

“DTI can also be useful for monitoring treatment reponse such as physical therapy and mental practice and white matter remodelling after stroke [39]. The combination of DTI, which evaluates the integrity of CST, and resting state-fMRI, which reflects brain connectivity, will provide more insight into the neuroplasticity mechanism after stroke [40,41].”

Reviewer 3 Report

The authors have used serial DTI to correlate imaging changes with clinical motor outcome.

This study is of potential interest.

Who adjudicated motor outcomes? Where they involved in the care of the patient or were they independent and blinded to DTI data.

Who evaluated the imaging outcomes? Did more than one radiologist read the studies? If discrepancies arose, how were these handled?

Author Response

Response to Reviewer 3 Comments

Thank you for giving me the opportunity to submit a revised draft of my manuscript titled ‘Prediction of Motor Recovery in Patients with Basal Ganglia Hemorrhage using Diffusion Tensor Imaging’ to journal of clinical medicine. We appreciate the time

and effort that you and the reviewers have dedicated to providing your valuable feedback on my manuscript. We are grateful to the reviewers for their insightful comments on my paper. We have been able to incorporate changes to reflect most of the suggestions provided by the reviewers. We have highlighted the changes within the manuscript.

Here is a point-by-point response to the reviewers’ comments and concerns.

The authors have used serial DTI to correlate imaging changes with clinical motor outcome.

 This study is of potential interest.

Point 1:  Who adjudicated motor outcomes? Where they involved in the care of the patient or were they independent and blinded to DTI data.

Response 1:  

Thanks for your thoughtful comments. The evaluation of motor outcome was conducted by a rehabilitation department and a resident who was trained professionally in evaluating the function of stroke patient recovery. All of them were not related to the study analysis, they were in charge of evaluation only and were blinded to DTI data.

Point 2:  Who evaluated the imaging outcomes? Did more than one radiologist read the studies? If discrepancies arose, how were these handled?

Response 2: Thanks for your thoughtful comments. The evaluation of imaging outcomes was independently performed by one radiologist and one rehabilitation physician with blindness. If there is discrepancy, it was evaluated again and averaged to calculate the final value.

Round 2

Reviewer 2 Report

I have following few minor issues with the revised manuscript: 

  • Although the authors responded in their response in reviewer’s comments file, but I believe that authors forgot to rephrase the term “value of diffusion tensor imaging” (e.g. in abstract line 34, Introduction line 95).
  • Lines 63-64 requires grammatical revision:

“ …new diffusion measures— quantitative anisotropy (QA) introduced to the field of….

  • Lines 72-73 requires grammatical revision:

“… ratio of bilateral hemispheres, and it can be compared with the previous studies which using FA value”.

  • Mean age is 50 (with SD 12), but authors mentioned that participants were 20 years of age and above. Does that mean that authors were unable to find participants with age between 20 and 38?

Authors mentioned that there was no sex differences: chi-square test (P = 1.000), but in the data authors have 8 males and 4 females. I am not sure how did the authors get p = 1 for sex differences.

Also, in the previous draft, authors mentioned that they had data from 7 males and 5 females, but in the current draft, they mentioned 8 males and 4 females.

  • I noticed that the voxels size of DTI data is highly anisotropic, which is usually not recommended for tractography. Did authors resampled the data before performing the analysis?
  • Authors mentioned that the color of fiber tracts indicated the two sides of hemisphere. This is usually not represented in most of the DTI studies. Mostly, fibers are color coded based on their directions e.g., right to left as ‘red’, anterior to posterior as ‘green’, and inferior to superior as ‘blue’. Authors indication of color codes might be confusing for the readers.
  • I am not sure why did the authors remove acknowledgment section?

Author Response

Response to Reviewer 2 Comments

We thank the reviewer for his/her attention and thoughtful second comments to our revised manuscript.

Thank you for giving me the second opportunity to submit a revised draft of my manuscript titled ‘Prediction of Motor Recovery in Patients with Basal Ganglia Hemorrhage using Diffusion Tensor Imaging’ to journal of clinical medicine.

I am really grateful by giving important and insightful comments about my paper. Here's a point-by-point response to the reviewer's 2nd comment and concern.

I have following few minor issues with the revised manuscript

Point 1 : Although the authors responded in their response in reviewer’s comments file, but I believe that authors forgot to rephrase the term “value of diffusion tensor imaging” (e.g. in abstract line 34, Introduction line 95).

Response 1 :  Thank you very much for your detailed review. As your thoughtful comment, "value of diffusion tensor imaging" was corrected to "usefulness of diffusion tensor imaging". The revised sentence is as follows.

This study aimed to investigate the usefulness of diffusion tensor imaging in predicting motor outcome after basal ganglia hemorrhage.

This study investigated the clinical usefulness of diffusion tensor imaging (DTI) in predicting the motor outcome in patients with basal ganglia hemorrhage.

Point 2 :  63-64 requires grammatical revision:

“ …new diffusion measures— quantitative anisotropy (QA) introduced to the field of….

Response 2 : Thank you very much for correcting all of your details. The grammatical revision was done as follows:

Recently, new diffusion measures—quantitative anisotropy (QA) were introduced in the DTI field for the analysis of diffusion properties of white matter.

Point 3 :  Lines 72-73 requires grammatical revision:

“… ratio of bilateral hemispheres, and it can be compared with the previous studies which using FA value”.

Response 3 : Thank you very much for your detailed review. In order to clarify the meaning of the sentence more, it was modified in two sentences. The revised sentence in manuscript is as follows:

Despite QA's methodological superiority, FA is the most basic parameter that is intuitively compared the degree of white matter integrity by comparing the ratio of bilateral hemispheres. Also, because the value of FA has been extensively used in previous studies, it is more appropriate to compare the results between studies.

Point 4 : Mean age is 50 (with SD 12), but authors mentioned that participants were 20 years of age and above. Does that mean that authors were unable to find participants with age between 20 and 38?

Response 4: According to the epidemiology of age-specific stroke incidence in Korea, the incidence of hemorrhagic stroke in younger ages under 44 is very low. Therefore, our study can reflect this and interpret that the proportion of young patients is relatively low.

Point 5 : Authors mentioned that there was no sex differences: chi-square test (P = 1.000), but in the data authors have 8 males and 4 females. I am not sure how did the authors get p = 1 for sex differences.

Respond 5 : Thank you very much for your careful attention to the qualification of our paper.

It was suggested that there was no difference in sex ratio between each group. Even in the good group, male : female = 4 : 2 vs poor group, male: female = 4: 2. Therefore ‘P = 1.000’ was found in the chi-squre test, and (P = 1.000) was also calculated in the fisher's exact test used when n number per group was expected to be less than 5.

Considering your concern, I revised it for clarity as follows:

There were no significant differences in age [P = 0.349, mann-whitney U] or sex ratio [P = 1.000, fisher’s exact test] between the good and poor outcome groups.

Point 6 : Also, in the previous draft, authors mentioned that they had data from 7 males and 5 females, but in the current draft, they mentioned 8 males and 4 females.

Response 6 :

Yes, you are right. Thank you so much for the opportunity to correct very critical content because of your detailed review. Again, this is our perfect mistake, thanks for your kind understanding.

Point 7 : I noticed that the voxels size of DTI data is highly anisotropic, which is usually not recommended for tractography. Did authors resampled the data before performing the analysis?

Response 7 : Thanks for your critical and thoughtful comments. Some studies have suggested that FA in CST is significantly affected by voxel size depending on the degree of crossing fiber. FA values ​​that are measured in regions containing crossing fibers are likely to be underestimate when using nonisotropic DTI. (https://pubmed.ncbi.nlm.nih.gov/17569968/, “Diffusion Anisotropy Measurement of Brain White Matter Is Affected by Voxel Size: Underestimation Occurs in Areas With Crossing Fibers”). I fully agree with your opinion. As you recommend, it may be better to isotropic the voxel size in future studies to qualify for tractography. The evaluation of imaging outcomes was independently performed by one radiologist and one rehabilitation physician with blindness. If there is discrepancy, it was evaluated again and averaged to calculate the final value.

Point 8 : Authors mentioned that the color of fiber tracts indicated the two sides of hemisphere. This is usually not represented in most of the DTI studies. Mostly, fibers are color coded based on their directions e.g., right to left as ‘red’, anterior to posterior as ‘green’, and inferior to superior as ‘blue’. Authors indication of color codes might be confusing for the readers.

Response 8 :

Thanks for your detailed review. As you recommended, to improve readability, pre and post 1,2,3 are not expressed in different colors as before, but all are unified in green and red colors as follows:

Figure 1. The corticospinal tract diffusion tensor tractography data in the good outcome group of three patients. The green color represents the corticospinal tract (CST) on the left hemisphere. Conversely, the red color represents the corticospinal tract (CST) on the right hemisphere.

Figure 2. The corticospinal tract diffusion tensor tractography data in the poor outcome group of three patients. The green color represents the corticospinal tract (CST) on the left hemisphere. Conversely, the red color represents the corticospinal tract (CST) on the right hemisphere.

Point 9 :  I am not sure why did the authors remove acknowledgment section?

Response 9 : This study was conducted with research funds in the Kyungpook National University Hospital. However, since the previously published journal, which was submitted at about the same time, remarked the acknowledgment, it was decided that the same acknowledgment would not have to be marked again in this paper. This is why we removed the acknowledgment remark.
